# Experiments on the Sinking of Marine Pipelines on Clayey Soils

Edgar Mendoza [1,*], María G. Neves [2], Cristina Afonso [3], Rodolfo Silva [1], André Ramos [2] and Miguel Losada [4]

1   Instituto de Ingeniería, Universidad Nacional Autónoma de México, Mexico City 04510, Mexico; rsilvac@iingen.unam.mx
2   Laboratório Nacional de Engenharia Civil (LNEC), 1700-066 Lisbon, Portugal; gneves@lnec.pt (M.G.N.); aramos83@gmail.com (A.R.)
3   Oceaning-Consulting Engineers, 1500-133 Lisbon, Portugal; cristina.afonso@oceaning.pt
4   Andalusian Institute for Earth System Research, University of Granada, 18006 Granada, Spain; mlosada@ugr.es
*   Correspondence: emendozab@iingen.unam.mx; Tel.: +52-55-5623-3600

**Abstract:** An experimental study was carried out to investigate seabed-pipeline interactions with regard to soil liquefaction. For a soil with a high proportion (30 to 60%) of fine sediment, four groups of tests were configured to reproduce soil liquefaction around pipelines for different initial pipe depths, pipe densities and wave conditions (wave height and period). The study focused on verifying the theoretically computed areas of soil failure by analyzing the sinking depths of the pipelines. The main findings are that a pipe with a submerged specific weight of less than half that of the soil will move up to the mudline; that the loss of soil loading capacity is more frequently evidenced in a fluid-like behavior of the soil than by an abrupt breaking of the soil matrix; and that the pipes which are totally buried will sink more than half-buried pipes. Moreover, wave action and the specific weight of the pipes seem to play more important roles in the expected behavior of the wave–soil–pipe interaction than the initial water content of the mud.

**Keywords:** wave-induced liquefaction; submarine pipelines; pipeline–seabed interaction; clayey soils

## 1. Introduction

Pipelines are used to transport hydrocarbon products or to transport and dispose of waste waters via a marine outfall. Research and development are of great importance in the former because the oil industry is still one of the main drivers of the global economy, and in the latter because the coastal population around the world continues to grow.

The good operational working of a marine outfall is extremely important for the environment, the welfare of the nearby population and the local economy. The structure must be safe and reliable throughout its lifetime. The two factors affecting this are variations in pipe strength and the mechanical behavior of the soil under wave action.

The behavior of submarine pipelines buried in loose granular soils [1] (e.g., silt, fine sand and, in some cases, gravel), or set on muddy soils is of great interest [2], especially the potential for pipeline floatation [3]. Previously, the damage related to the displacement and sinking of pipelines was thought to be mainly due to geotechnical failures, such as soil liquefaction, soil fluidization and loss of the soil's load capacity [4]. Failure of submarine pipelines has been linked to the wave-induced instability of marine deposits, which can lead to liquefaction [2,5,6].

In recent decades the understanding of hydrodynamics and erosion around marine structures has improved, although there is still little known about the impact of soil liquefaction on these structures [7]. Sumer, 2014 [8] found that, as a result of soil liquefaction,

buried pipelines may float to the surface of the seabed, and pipelines laid on the seabed may sink to greater depths.

Soil liquefaction caused by earthquakes has been studied to some extent in the past thirty years [9]. However, much more attention, comparatively, has been paid to wave-induced liquefaction [10,11]. Indeed, while the design of marine structures has substantially advanced of late, research into the design of their foundations, with regard to soil liquefaction, has received little attention [8].

From 2001 to 2004 the EU funded a research project called LIMAS (Liquefaction around Marine Structures), and important steps were taken in the study of failure modes of marine structures, including submarine pipelines, due to wave-induced liquefaction [12]. This project was preceded by another EU research program (1997–2000) on scour around coastal structures (SCARCOST) in which liquefaction around coastal structures was one of the focus areas [8].

Within the scope of the LIMAS project, laboratory tests were conducted, focused on understanding the processes that occur during soil liquefaction around pipes, due to wave action [7,12–14]. Before that, Sumer et al., 1999 [15] carried out an experimental study on the sinking of marine objects, such as pipelines, free to move vertically, on a seabed exposed to progressive waves that became liquefied due to the build-up of pore-water pressure. Similar tests were undertaken [13,14], but with a pipe moving vertically and horizontally, measuring the pore-water pressure across the soil depth and the pipeline displacement. In both studies, the results showed that the pipeline adjusted its position in the liquefied bed depending on the ratio of the specific gravity of the pipeline, $s_p$, and the specific gravity of the liquefied soil. Sumer et al., 2006 [12] carried out experimental work on pipeline floatation. They concluded that the pipeline remained where it was when its density was equal to the density of the surrounding liquefied soil, or its motion stopped at the point where these densities were equal. Sumer et al., 2006 [7] also carried out tests on liquefaction around a buried pipeline in a soil that was exposed to progressive waves. They analyzed the influence of the pipe on the liquefaction of the soil around it, with the pressure building-up more rapidly at the bottom of the pipe, than farther away, at the same depth below the water soil interface. At the top of the pipe, the excess pore pressure is not affected by the presence of the pipe.

Based on these test results, Teh et al., 2006 [13] developed and validated a method to determine the minimum specific gravity for a pipeline to become self-buried, by taking into consideration the density of liquefied soil. In the study, only one type of soil was tested, and the researchers also proposed changes in pipeline design, which took into account the possibility of soil liquefaction [13].

Zhao et al., 2018 [2] used the results from Sumer et al., 2006 [7], to calibrate a simple but accurate numerical model. This is based on the Biot theory and simulates wave-induced liquefaction around a pipeline in a sandy seabed. The model incorporates features such as the relation for the consolidation, which describes the pore-volume reduction, nonlinear stress-strain behavior of the soil skeleton and the effect of soil–pipe contact. The numerical results obtained show good agreement with the laboratory results of Sumer et al. [7], assuring the reliability of the method for the prediction of wave-induced liquefaction in a sandy seabed and the resulting failure of a buried pipeline.

However, these tests were undertaken mainly for a soil composed of sand or silt and it was found that the soil composition significantly changes the resistance of the seabed to erosion [6]. According to Sumer, 2014 [3], wave-induced seabed liquefaction around marine structures is mostly limited to fine soils, such as silt or fine sands, with grain sizes d = 0.074–0.4 mm, or composite soils, such as silty sands or clayey sands, with grain sizes d = < 0.074 mm. For pipelines laid on muddy soils, there is still great uncertainty, especially related to settlement, scour and liquefaction.

An experimental study on soil-wave interaction in the presence of pipelines using a clayey soil (with a clay content of 57%), was conducted by Vijaya kumar et al., 2005 [16].

The present tests therefore give one of the greatest set of results for clay content available in the literature.

In that study, wave pressures on submarine pipelines due to waves were measured for pipelines at three seabed depths and for different soil compositions, to investigate the influence of wave- and pipeline-related parameters. These authors concluded that the uplift forces are smaller when the pipeline is fully buried, than when the pipeline is resting on the seabed. Based on that data, Postacchini and Brocchini, 2015 [17] analyzed the main parameters which characterize the scouring process in a cohesive soil. They concluded that the scour depth under a pipeline lying on a cohesive soil depends only on the Keulegan–Carpenter parameter (directly) and on the clay content (inversely). They found that when the clay content is higher, the scour depth depends directly on both these parameters.

This study aims to understand the impact of soil liquefaction on submarine pipelines, for a soil with a high content of fine sediment. The experiments presented here investigate wave–seabed–pipeline interaction, focusing on the role of the initial water content of the soil, the relative pipe densities and the initial position of the pipelines. For the purposes of this work, and following the studies in [18,19], soil liquefaction is defined as the physical process of the loss of bearing capacity of the soil, as the aggregate of particles loses its resistance, due to the cyclic action of the oscillatory regime. Fluidization refers to one particular case of liquefaction, more common in cohesive soft soils. When waves propagate into the bed, the aggregate of particles is suspended in a fluid matrix that is destroyed by the accumulation of pore pressure. The soil behaves as a fluid and causes seabed slide. In turn, excess pore pressure is the increase in water pressure, above the static value, due to waves.

The main objective of this paper was to analyze the soil–pipe–wave interaction, specifically in the parts of the soil in which the theoretical excess pore pressure was greater than the initial mean normal effective stress. In addition, the analysis of the loss of the soil's load capacity and the behavior of submarine outfalls on muddy soil were also examined.

The tests were carried out in the facilities described by Chávez et al., 2017 [20] and Section 2 describes the experimental set up used to reproduce soil liquefaction around pipelines for different initial pipe positions, pipe density and wave conditions. The experimental methodology implemented is also given. Section 3 includes the results of each test, organized into discussion groups: pore pressure distribution in time, wave height evolution versus vertical displacement of the pipes and maximum pore pressure distribution.

## 2. Materials and Methods

The experiments were carried out at the laboratory of the Engineering Institute at the National Autonomous University of Mexico. There, the wave flume is 22 m long, 0.4 m wide and 0.6 m deep and the waves are generated by a piston-type wave maker with an active absorption system. A gravel dissipative beach was constructed at the end of the flume in order to absorb undesired reflection.

At the bottom of the flume, there is a removable section (13 m from the wave maker), that was replaced for these tests by an acrylic pit, 0.84 m long, 0.285 m wide, and 0.2 m deep (see Figure 1). The area of clayey soil was 2.44 m in length, including the pit and the space before and after it, between the two ramps (black triangles in Figure 1). Outside the pit, the clayey soil had a thickness of 0.05 m. The water depth outside the experimental area was 0.3 m, while in the experimental area it was h = 0.25 m.

The pore pressure of the soil in the acrylic pit was recorded by 28 pressure transmitters (0–0.2 bar Acculevel, Keller, Minneapolis, MN, USA). The transducers were set in a matrix of seven rows by four columns, as shown in Figure 1. The pressure sensors were located at 9, 13, 17 and 21 cm from the soil-water surface and at 0, 12, 24, 42, 60, 72 and 84 cm from the beginning of the soil pit.

Eleven wave gauges were placed along the flume so that (1) the incident and reflected waves could be separated and (2) the wave evolution over the physical model could be retrieved. Only wave gauges 04 to 08 were placed in the experimental area (see Figure 1).

The muddy soil section was extended 0.8 m before and after the pit, to avoid contaminating the soil response in the pit due to the sudden change in the flume bottom, as well as to steepen the waves and, consequently, to increase the wave-induced stresses. The ramps used to confine these two areas of muddy soil were made of acrylic.

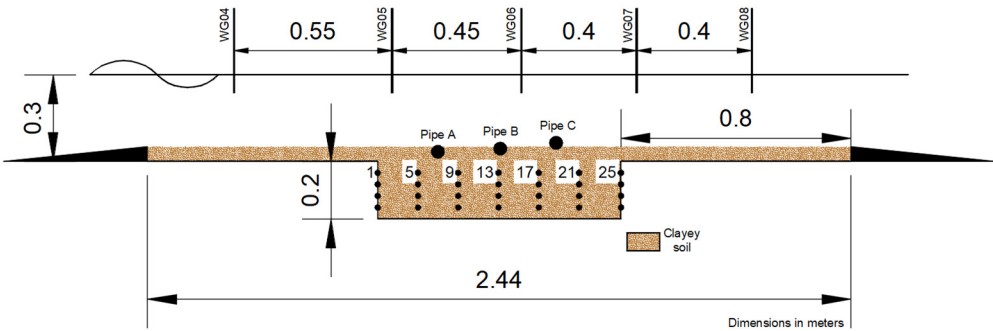

**Figure 1.** Experimental set-up with detail of the acrylic pit, showing the pressure transducers (series of black dots) and the wave gauges (WG) above.

The soil used in the tests was made "in-house", using a mixture of commercial kaolinite and sand, brought from the coast of Puerto Morelos, Mexico ($D_{50}$ of 0.267 mm). Commercial kaolin has been successfully tested previously [21,22]. Five different types of soil were tested preliminarily, ranging from all sand, identified as 100A, to all mud, identified as 100C. Between these extremes, three combinations of sand (A) and mud (C) were tested: 85% sand—15% kaolinite (85A15C); 60% sand—40% kaolinite (60A40C); and 30% sand—70% kaolinite (30A70C). The physical and mechanical properties of the soil, measured in the laboratory, are shown in Table 1, where $e_{max}$ and $e_{min}$ are the maximum and minimum void ratios (determined in the laboratory), $\rho_s$ is the soil mass density, $\rho_w$ is the water mass density and $D_{50}$ is the value of the particle diameter at 50% in the cumulative distribution.

**Table 1.** Physical and mechanical properties of the soil mixtures.

|          | $\rho_s/\rho_w$ | $D_{50}$ (mm) | $e_{max}$ | $e_{min}$ |
|----------|-----------------|---------------|-----------|-----------|
| 100A     | 2.85            | 0.267         | 1.845     | 0.992     |
| 85A15C   | 2.82            | 0.245         | 1.551     | 0.795     |
| 60A40C   | 2.73            | 0.180         | 1.998     | 0.726     |
| 30A70C   | 2.60            | 0.070         | 2.383     | 0.995     |
| 100C     | 2.17            | 0.002         | 3.032     | 1.100     |

The resulting particle size distribution for each mixture is shown in Figure 2.

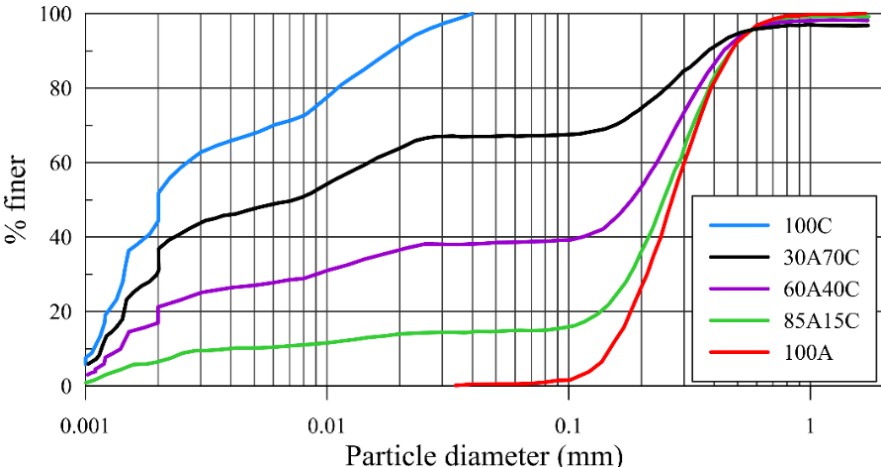

**Figure 2.** Granulometric curve of the soils used in the preliminary tests.

According to the Unified Soil Classification System (USCS), taking into account the Atterberg limits determined in the laboratory (liquid, LL = 34.20, and plastic, LP = 26.15) and the plasticity index, IP = 8.05, the kaolinite used in the tests is a clay with low plasticity (CL). After preliminary testing and soil characteristic analysis, which is fully described by Chávez et al. [20], the soil mixture 85A15C was selected. This soil was produced with a solid-water ratio of 2.0 kg/L. The mixture was selected as it has sufficient load bearing capacity but, under certain wave conditions, liquefaction is possible. This means that if the pipes sank during wave action it would be mainly due to soil liquefaction and not due to the weight of the pipes. The PVC pipes used in the experiments were 3 cm in diameter and filled with sand to get the corresponding submerged specific weight.

Four groups of tests were performed (C1 to C4) with regular incident waves, as shown in Table 2. Each group consisted of four wave conditions (WT1 to WT4) and the duration of each condition was 1200 s. The test conditions corresponded to intermediate water depth, similar to those of existing submarine outfalls at sites such as Portugal, Morocco and Malta.

**Table 2.** Set up for each wave condition (WT); initial position and submerged specific weight of the pipes for each group of tests.

| Test | WT | $T$ (s) | $H$ (m) | Relative Initial Position [1], $z_i$ | | | $s_p$ | | |
|------|-----|---------|---------|--------|--------|--------|--------|--------|--------|
| | | | | Pipe A | Pipe B | Pipe C | Pipe A | Pipe B | Pipe C |
| | 1 | 1.00 | 0.12 | | | | | | |
| C1 | 2 | 1.00 | 0.15 | 1.16 | 0.66 | 0.33 | 1.80 | 1.80 | 1.80 |
| | 3 | 1.20 | 0.12 | | | | | | |
| | 4 | 1.20 | 0.15 | | | | | | |
| | 1 | 1.00 | 0.12 | | | | | | |
| C2 | 2 | 1.00 | 0.15 | 1.23 | 1.13 | 1.00 | 2.00 | 1.80 | 1.60 |
| | 3 | 1.20 | 0.12 | | | | | | |
| | 4 | 1.20 | 0.15 | | | | | | |
| | 1 | 1.00 | 0.12 | | | | | | |
| C3 | 2 | 1.00 | 0.15 | 0.55 | 0.55 | 0.55 | 2.00 | 1.80 | 1.60 |
| | 3 | 1.20 | 0.12 | | | | | | |
| | 4 | 1.20 | 0.15 | | | | | | |
| | 1 | 1.00 | 0.12 | | | | | | |
| C4 | 2 | 1.00 | 0.15 | 1.66 | 1.66 | 1.66 | 1.20 | 1.30 | 1.60 |
| | 3 | 1.20 | 0.12 | | | | | | |
| | 4 | 1.20 | 0.15 | | | | | | |

[1] $z_i = z/\varnothing$, where z is the position below the mudline and $\varnothing$ the pipe diameter (3 cm).

The values of $s_p$ (submerged specific dead weight of the pipe) used in the experimental tests, namely 1.2 to 2, are similar to those of existing submarine outfalls, such as those at Lagoa-Meco, Portugal [23], Ta'Barkat, Malta [24], and Tanger and Tetuan, both in Morocco [25].

Each group of tests differed in the initial burial depth of the three pipes in the clayey soil, and in the submerged specific weights (see Table 2). The position of the pipes, z, is the distance from the pipe to the soil surface, i.e., zero would correspond to a pipe resting on the bed and 3.0 cm to a totally buried pipe. In Table 2, the initial positions are shown relative to the pipe diameter. The pipes were referred to as Pipe A, closest to the wave-maker, 92 cm away from the beginning of the clayey soil area, Pipe B, placed 15 cm from Pipe A, and Pipe C, located 33 cm further away. These distances were set to ensure their visibility in the tests (see Figure 3). The direction of the wave propagation was from Pipe A towards Pipe C. The PVC pipes were placed inside plastic rails to allow only vertical displacement (see Figure 3). The rails were slightly wider than the pipe diameter to minimize friction effects during the displacement of the pipes. Given that the pipes were placed on the soil, and in some cases buried, the hydrodynamic alteration due to the pipes themselves, or the recording equipment, is negligible. The same pipes were used in all the tests, thus

the interaction between the soil and the pipe roughness is a constant and its analysis falls beyond the scope of this research.

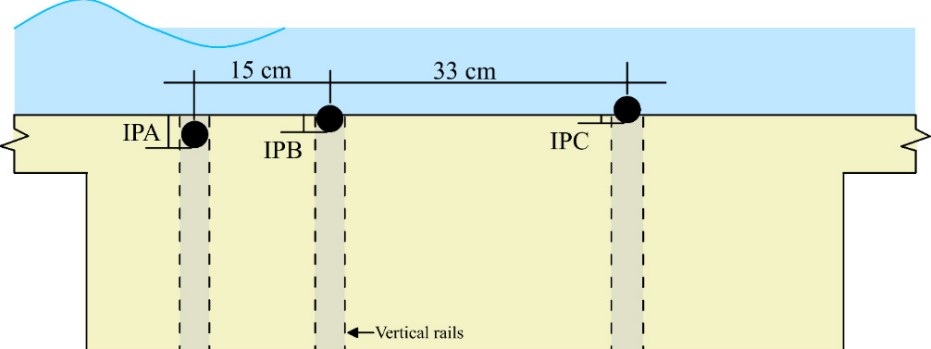

IPA= Initial postion of pipe A; IPB= Initial postion of pipe B; IPC= Initial postion of pipe C

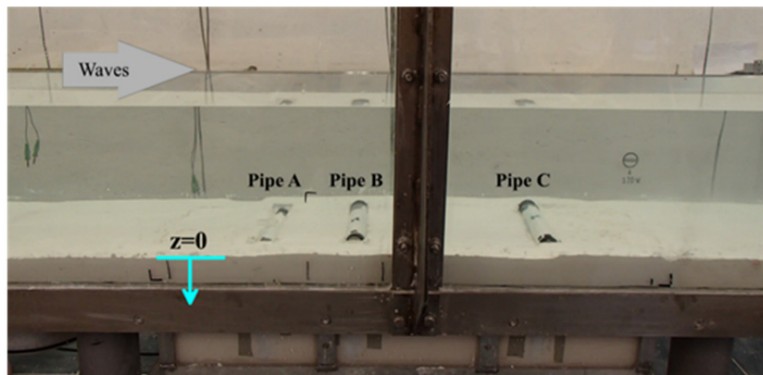

**Figure 3.** Sketch of the experimental area (**top**), and photograph showing the pipes in position, in the wave flume (**bottom**).

The experimental procedure is similar to that presented by Chávez et al., (2017) [20]:

1. Prepare the mud by rubbing it by hand until a homogenous mixture is obtained;
2. Fill the pit with the mud to the top level of the ramps;
3. Let the soil consolidate until it is able to bear the structure weight (typically a few hours);
4. Fill the flume with 0.30 m of water, deploy the three pipes and allow the system to consolidate for 24 h;
5. Calibrate the wave gauges and set the pressure transmitters to zero, to directly record the excess pore pressure;
6. Switch on the measurement system and start the waves.

After each wave condition, the pipes were replaced at their initial positions. The clayey soil was renewed after each group of tests. Then, steps 4 to 6 were repeated.

In general, the tests carried out were within the range of values of the main dimensionless parameters presented in the literature, listed in Table 3, for tests of pipes in clayey soils.

In Table 3, $Re$ is the Reynolds' number, $KC$ is the Keulegan–Carpenter number, $D$ is the pipe diameter, $h$ is the water depth, $H$ is the wave height, $L$ is the wave length and $d$ is the thickness of the soil layer. All of these parameters were determined following Sumer et al., 2006 [12].

The main differences between previous and the present tests are: (1) the $D_{50}/L$ of the present tests was slightly higher than the values reported in the literature; and (2) the type of soil in most of the previous tests was composed of sand or silt, except in [16], where clay was used. This means that the results and findings of the present work apply to soil with a high content of mud, which had not been studied before.

**Table 3.** Dimensionless parameters for the present tests and for previous tests.

| Dimensionless Parameter | Previous Tests [7,12–16] | | Present Tests |
|---|---|---|---|
| | Min | Max | |
| $Re$ | $1.08 \times 10^4$ | $6.52 \times 10^6$ | $1.06 \times 10^4$ |
| $KC$ | $1.01 \times 10^0$ | $3.69 \times 10^3$ | $1.59 \times 10^1$ |
| $h/L$ | $9.05 \times 10^{-2}$ | $3.31 \times 10^{-1}$ | $1.46 \times 10^{-1}$ |
| $H/L$ | $1.23 \times 10^{-2}$ | $5.89 \times 10^0$ | $5.35 \times 10^{-2}$ |
| $D_{50}/L$ | $1.56 \times 10^{-5}$ | $4.45 \times 10^{-5}$ | $3.16 \times 10^{-4}$ |
| $H/D$ | $3.13 \times 10^{-1}$ | $8.50 \times 10^2$ | $3.67 \times 10^0$ |
| $d/L$ | $5.89 \times 10^{-2}$ | $4.37 \times 10^{-1}$ | $9.72 \times 10^{-2}$ |
| $D/h$ | $4.76 \times 10^{-2}$ | $5.33 \times 10^{-1}$ | $1.00 \times 10^{-1}$ |
| $D/d$ | $1.14 \times 10^{-1}$ | $4.71 \times 10^{-1}$ | $1.50 \times 10^{-1}$ |

## 3. Results

Given that the critical data recorded during the experiments is the pore pressure within the soil, Figure 4 shows an example of the time series of period averaged pressures recorded. The plots correspond to C1 tests. It can be seen that pore-water pressure exceeding the static pore-water pressure, $p$, is dimensionless by the product of the specific weight of water, $\gamma$ and the water depth, $h$. Figure 4 shows the time series corresponding to the upper line of pressure transmitters, namely sensors 1, 5, 9, 13, 17, 21 and 25 (see Figure 1). These sensors were placed closest to the mudline, 9 cm below the soil-water interface, and thus show the highest excess pressures.

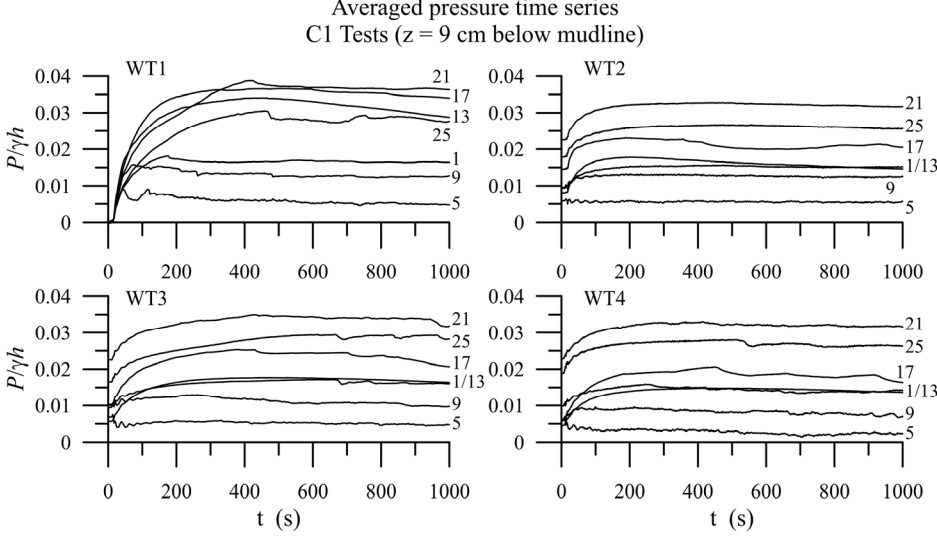

**Figure 4.** Phase averaged pressure time series for the wave conditions in tests C1.

In general, all the time series of Figure 4 show a building-up of pressure, reaching maximum values after around 200 s of wave action. After that, the pressure did not alter radically. During the tests, it was noticeable that there was no sudden pressure build-up nor abrupt pressure decrease, which means that the soil matrix was not destroyed. Nevertheless, fluid-like behavior was observed and a partial loss in loading capacity was evidenced by the pipe sinking. The results shown are a representative example of the pore pressure behavior in all the experiments. Complementary data can be found in Appendix A.

In the following sections the results of the experiments and the main findings, related to three aspects, are presented: (a) the wave height distribution along the clayey soil; (b) the estimation of the theoretical liquefaction zone within the pit; and (c) the vertical displacement of the pipes.

### 3.1. C1 Tests

As mentioned earlier, a 5 cm layer of mud was placed before and after the acrylic pit to get higher waves and to concentrate the stresses on the soil. Figure 5 shows the distribution of the significant wave heights (Hs) recorded along the clayey soil, for C1 tests. It can be seen that wave steepening occurred gradually and the maximum wave height was reached around the middle of the pit. After that, the wave height fell rapidly. No wave breaking occurred. The location of the maximum wave height was found where the soil is expected to fail.

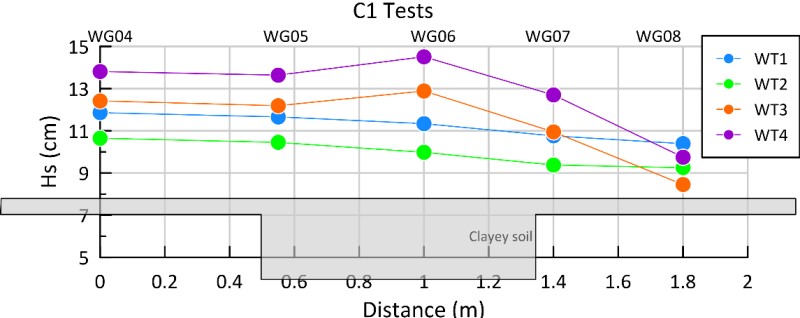

**Figure 5.** Significant wave height distribution along the clayey soil in C1 tests. The shaded area (clayey soil) is shown for reference.

In order to analyze the maximum pore pressures recorded during the experiments, and to relate this with the areas of theoretical soil failure, pressure maps were plotted showing the instant of maximum pressure for each wave condition, shown in Figure 6. For the instant of maximum pressure measured by each transmitter, a linear interpolation was performed, obtaining a pressure map for the whole area. Then, the initial mean normal effective stress was estimated, using equation 1 [26] for the whole soil area, and the zones where this value was exceeded were added to the maximum pressure map. In Equation (1), $\sigma'_0$ is the initial mean normal effective stress, $\gamma'$ is the submerged specific weight of the soil, $k_0$ is the coefficient of lateral earth pressure, as given by [20] and is equal to 0.441; $z$ is the vertical coordinate measured downwards from the upper surface of the soil to the bottom of the flume.

$$\sigma'_0 = \gamma'z(1 + 2k_0)/3 \tag{1}$$

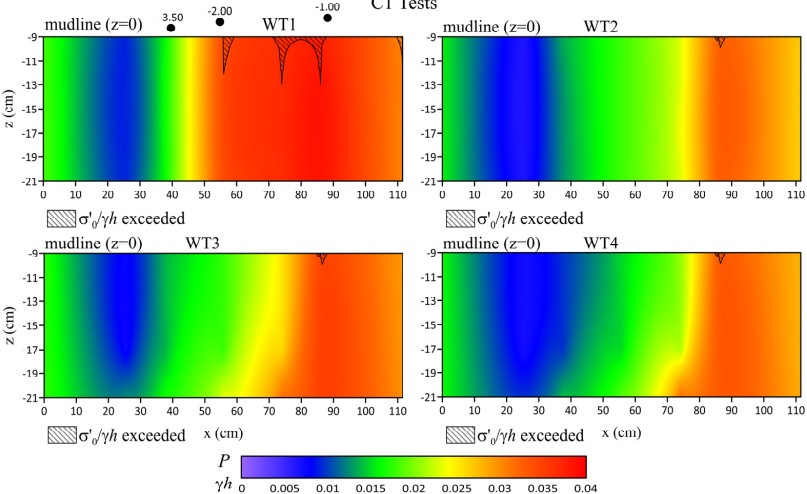

**Figure 6.** Contour maps of maximum pressures and areas where the initial mean normal effective stress was exceeded, for C1 tests. The initial position of the pipes is shown in the upper left panel. The horizontal axis is the length of the pit.

For the soil used in this study, Equation (1) yields the linear relation $\frac{\sigma'_0}{\gamma'h} = 0.228z$, where $z$ is in meters. This indicates that the greater the depth, the greater the pressure needed to produce soil liquefaction.

From Figure 6 it can be seen that the largest areas of soil failure, where initial mean normal effective stress was exceeded, were produced with wave condition WT1. Other wave conditions produced areas that were negligible in size. This may be due to use of the same soil in WT1 and WT2 tests, which produced cumulative effects and/or a greater pore pressure build-up with short waves than that produced by longer waves. The main characteristic observed in the pressure maps is that the horizontal variation of the pressure value increases in the direction of the wave propagation. A less significant variation is seen in the vertical direction, causing the soil to retain its loading capacity and limiting the depth to which the pipes sink. Time series of all C1 Tests 17 cm into the soil are shown in Appendix A (Figures A1–A7).

To analyze the relationship between the spatial evolution of the wave height and the vertical displacement of the pipes, Figure 7 shows the vertical displacement for each WT. The sinking distances, $Dz$ = final position − initial position, of pipes A, B and C were plotted, as well as the maximum pressures recorded by the transmitters located closest to the clay-water interface.

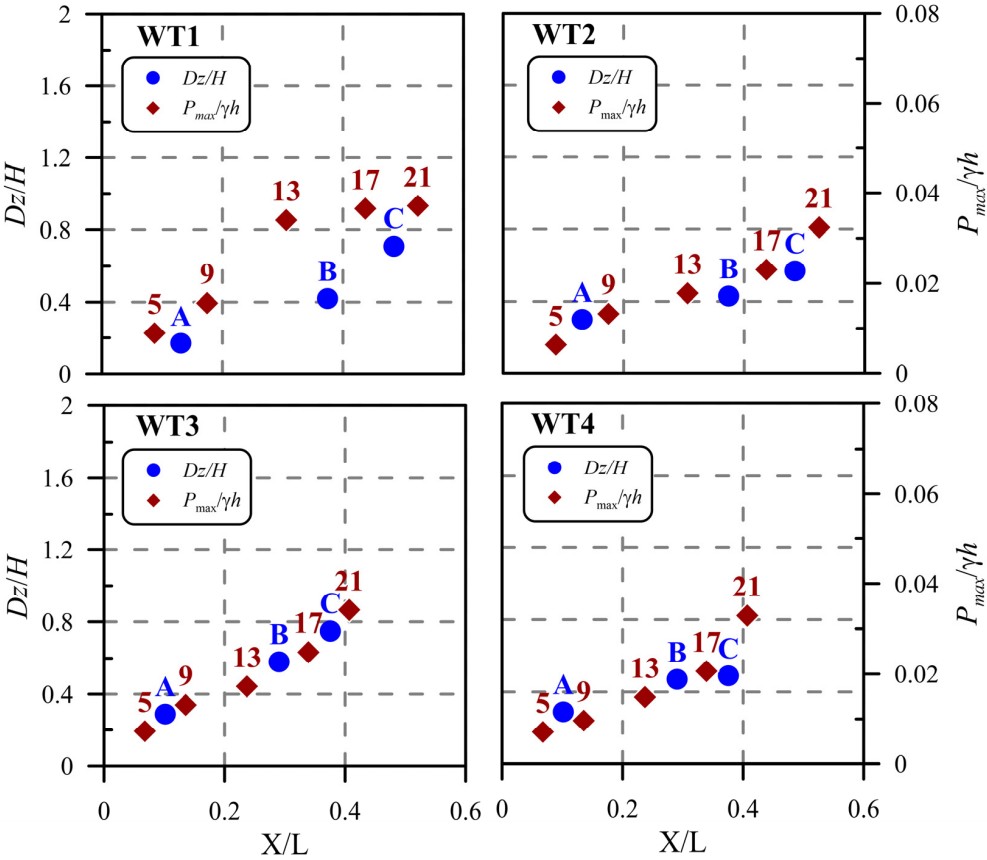

**Figure 7.** Vertical net displacements (final position–initial position) for pipes A, B and C (blue dots) and maximum pressures recorded by the transmitters, 9 cm below the mudline for all C1 tests.

In all the wave conditions of the C1 tests, the maximum vertical displacement was for pipe C (8–10 cm), followed by pipe B (5–7 cm), both located close to the zone where the initial mean normal effective stress was exceeded. Pipe A showed least displacement (2–5 cm). This may be due to the variation in the pressure distribution in the flume, since the pressure is greater for the areas in the acrylic pit farther from the wave generator (pressure

transmitters 17 and 21), meaning that the pipe buried at the shallowest depth is the pipe which sank most.

This result is related specifically to the soil behavior, given that all the pipes had the same submerged specific weight and, as mentioned earlier, the four WT in group C1 show very similar responses in soil fluidization, pressure distribution and pipe response. Moreover, the initial position changes from pipe to pipe, with pipe A being the only one totally buried, having less wave-soil interaction and the pipe which sank least. Pipe C sank most and was initially the least deeply buried. These results seem to indicate that if the pipe is not totally buried, the wave-soil-pipe interaction could lead to the pipe being buried deeper than if it had been totally buried. This is in agreement with the results of Vijaya kumar et al. [16] but not with those of Teh et al. [14], who stated that for cases with different wave heights, but the same periods, the initial position is not a determining factor in the final embedment of the pipe.

*3.2. C2 Tests*

In none of the C2 tests was a pressure build-up process recorded. This means that the water within the clayey soil was able to drain freely, as in non-confined conditions; this may be due to the natural inhomogeneity of the soil, or that induced in the making of it. In these tests, all the pipes were totally buried initially, but with different embedment depths and different submerged specific weights. Figure 8 shows the wave height-distribution along the soil. It can be seen that the wave heights did not increase, instead the wave heights decreased, which may be the reason for the low stress states recorded.

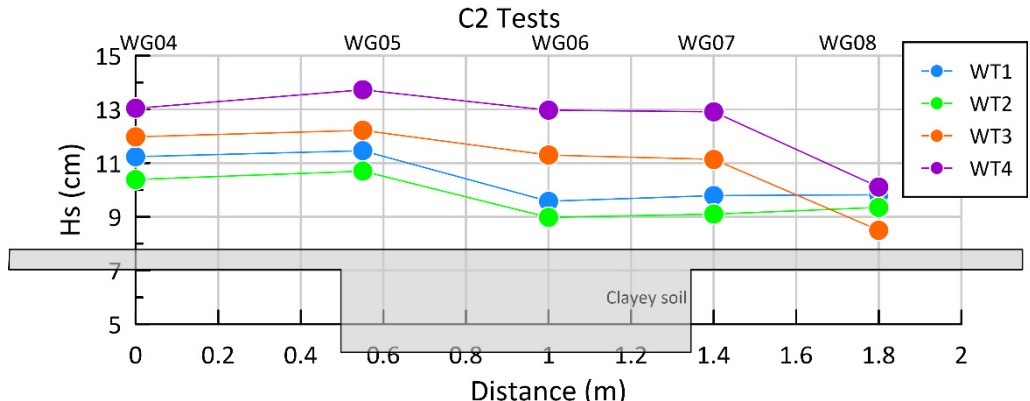

**Figure 8.** Significant wave height distribution along the clayey soil in C2 tests. The shaded area (clayey soil) is shown for reference.

Interestingly, although no abrupt soil failure occurred in the C2 tests, fluid-like behavior was observed, and the pipes sank during the wave action. This is seen in Figure 9, where the maximum pressure and the vertical displacement of the pipes are shown. Even with the low stresses found, the net displacements of the pipes were more than 0.08 m, that is, greater than those in tests C1. In the C2 tests, the greatest displacement was for pipe B (12–16 cm), then pipe C (8–14 cm) and lastly, pipe A (8–11 cm), for both WT1 and WT2. Pipes A and C had similar displacements for WT3 and WT4. In these tests, neither pipe C (closest to the mudline and with the lowest specific weight), nor pipe A (with greatest specific weight and initially buried nearer the surface), sank further in the four tests. Arguably, this is evidence of soil anisotropy, given that the sinking of the pipes is due to the loss of soil loading capacity and that the combination of the initial position and the specific weight of the pipes are relatively high in importance.

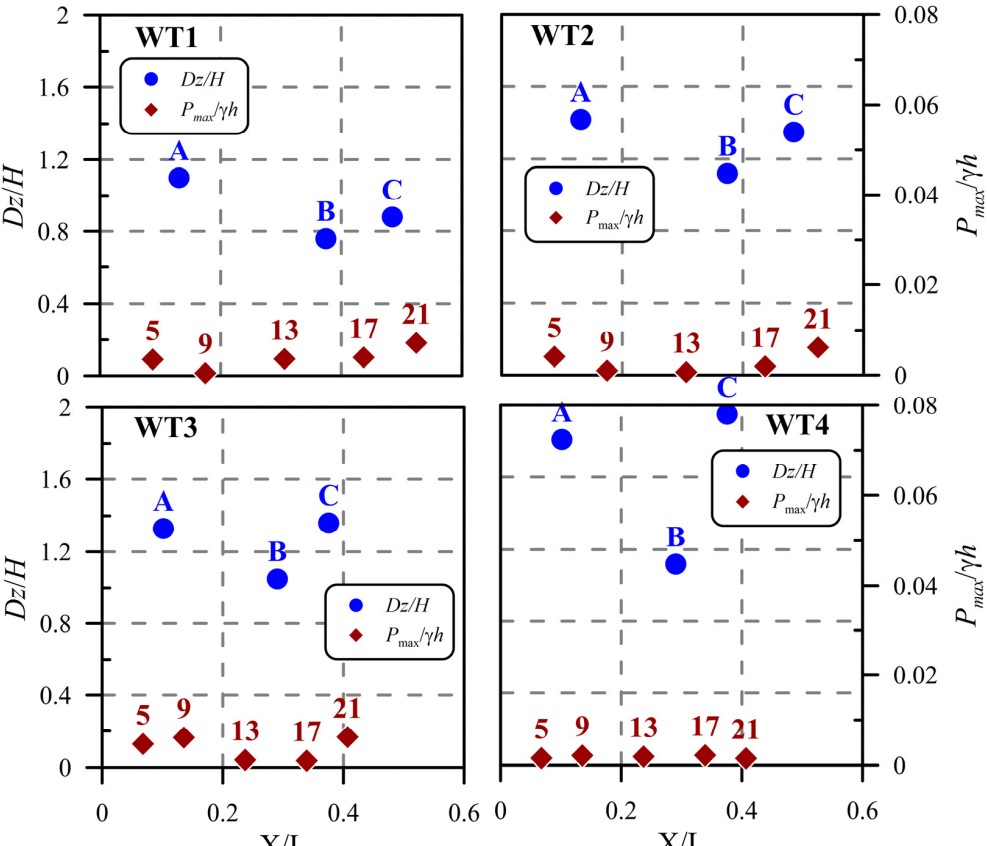

**Figure 9.** Vertical net displacements (final position–initial position) for pipes A, B and C (blue dots) and maximum pressure recorded by the transmitters in the line 9 cm below the mudline for all C2 tests.

Furthermore, the pipe with the greatest specific weight, that was initially buried lowest, pipe A, showed least displacement in most of the wave tests, and the distribution of pressure values is almost homogeneous all along the line 9 cm below the mudline, never exceeding 0.025 m.

### 3.3. C3 Tests

The pressure time series of the wave conditions in this group, where the three pipes were almost half-buried and have different specific weights, show very different results from pipe to pipe, as can be seen in Figure 10. The only common response in the pressure recordings is the build-up process, which reaches higher values in WT1 and WT3 then in conditions 2 and 4. Apart from that, it is difficult to find similarities, unlike the C2 tests, even though these only differ from the C3 tests in the initial position of the pipes.

In WT1 the transmitters farthest from the wave maker (17 and 21) show a rapid increase in pressure (0 to 300 s) and a slow pressure release for the rest of the test; sensors 9, 13 and 25 show a continuous increase in the pressure for the entire period of the test. In turn, transmitters 1 and 9 show increasing pressure at the beginning, followed by release at end of the tests. For WT2 all the time series show a small pressure increase followed by a slow release of pressure. In WT3 the time series behavior is similar to that of WT1, but with smaller variations over time. In WT4 only the pressure in transmitter 21 increases significantly throughout the test, similar to a transitional liquefaction [15]. Complementary data can be found in Appendix A.

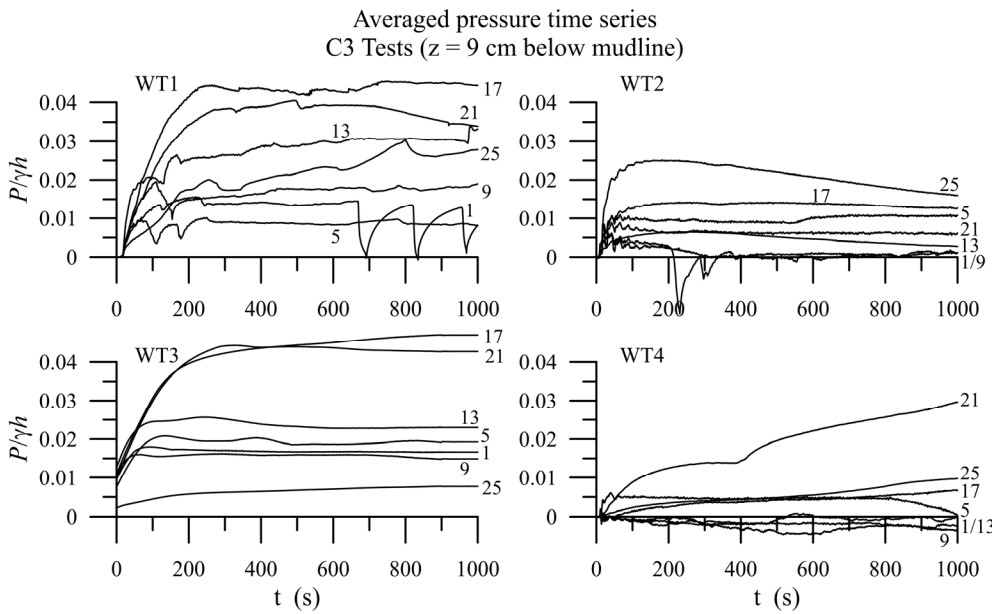

**Figure 10.** Phase averaged pressure time series for the wave conditions of C3 tests.

The significant wave height distribution is shown in Figure 11. Again, no steepening occurred and a decrease in wave height was recorded in the wave direction. For all C3 tests a noticeable decrease in wave height is seen close to the middle of the acrylic pit, after this a small recovery occurred, to decay again towards the end of the clayey soil area.

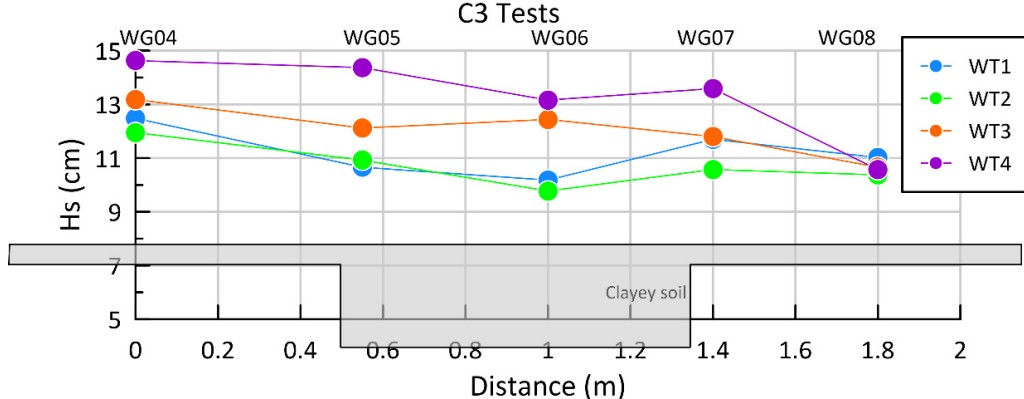

**Figure 11.** Significant wave height distribution along the clayey soil in C3 tests. The shaded area (clayey soil) is shown for reference.

Figure 12 shows the maps of maximum pressure for the C3 tests with the areas of theoretical failure estimated by equation 1. This group has the most extensive areas of computed failure, specifically for WT1 and WT2. The horizontal pressure variations are noticeably greater than the vertical and, as in the C1 tests, the highest pressures occur in the area of the pit farthest from the wave maker. This pressure distribution, found repeatedly, suggests that the wave-induced stresses are exerted on the soil depending on how the wave propagates over the soil. It is unlikely that the pressure distribution is an effect of the confinement of the pit, as the pressures recorded at the very back of the pit are not the highest. Time series of all C3 Tests 17 cm into the soil are shown in Appendix A (Figures A8–A14).

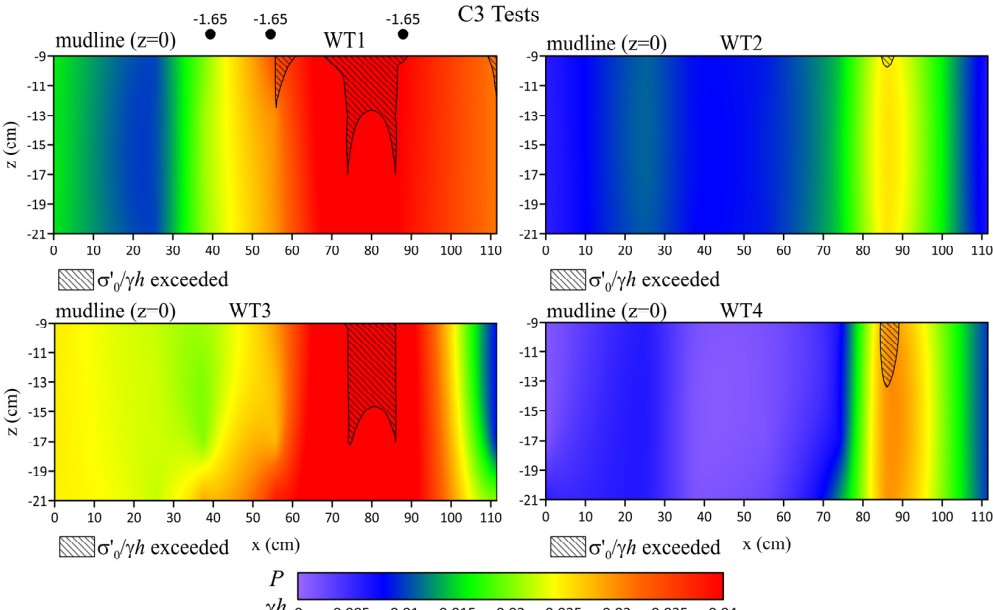

**Figure 12.** Contour maps of maximum pressure and areas where the initial mean normal effective stress was exceeded, for C3 tests. The initial position of the pipes is shown in the upper left panel. The horizontal axis is the length along the pit.

Figure 13 also shows that the soil failure in the C3 tests occurred at the greatest depths of all the tests. As a result, the vertical displacements of the pipes in these tests are also the greatest of all the tests in this research.

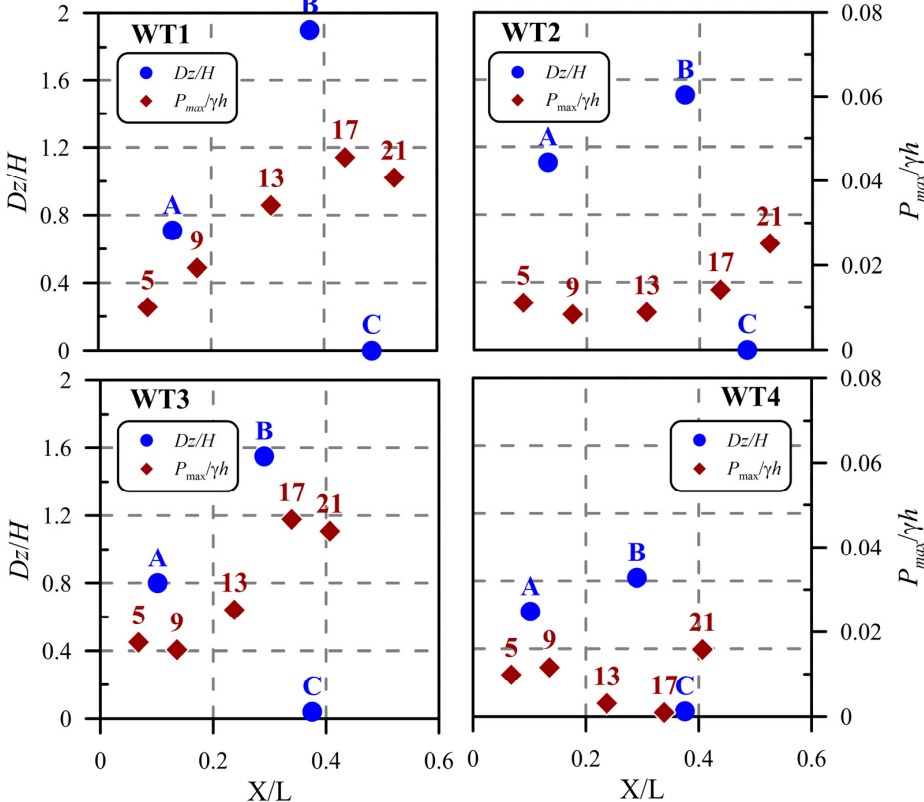

**Figure 13.** Vertical net displacements (final position–initial position) for pipes A, B and C (blue dots) and for all C3 tests, maximum pressure recorded by the transmitters in the line 9 cm below the mudline.

In Figure 13 it can be seen that pipe B moved down from 11 to 23 cm; pipe A had a net sinking of 9 to 17 cm and pipe C kept its initial position. The behavior of pipes A and B is as expected, given their specific weight; while pipe C should have also have sunk, but did not. This is interesting, considering that this pipe was located in the area where maximum pressures occurred.

An explanation for the behavior of pipe C is that, although the depth at which the soil fluidized was less than the depth of the pipe, the pore-water was able to drain and the area surrounding the pipe retained enough loading capacity to keep the pipe in place. This occurred in combination with the low submerged specific weight of the pipe.

*3.4. C4 Tests*

In the C4 tests, with all the pipes totally buried and with the lowest specific weights tested, the most irregular pressure responses were found. This could be due to the irregular wave height behavior, seen in Figure 14. In this figure, in the tests with the shorter wave period (WT1 and WT2) an abrupt wave height decay was seen close to the middle of the pit; while in tests WT3 and WT4 the wave height decay is gradual.

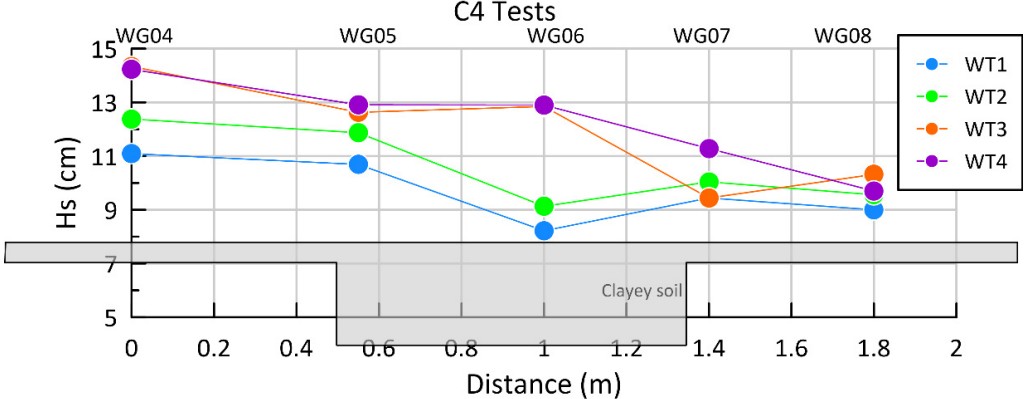

**Figure 14.** Significant wave height distribution along the clayey soil in C4 tests. The shaded area (clayey soil) is shown for reference.

Figure 15 shows the vertical displacements of the pipes and the maximum recorded pressure for the C4 tests. Pipes A and B had the lowest specific weights. These pipes were not drawn in Figure 15 because both moved up and were found at the mudline after the wave condition was ended. For WT1 the highest pressure was recorded by transmitter 9, so it would seem likely that this area would lose loading capacity. However, the buoyancy force was greater here than the weight of pipe A and therefore it rose towards the water soil interface. This result suggests a preliminary conclusion on pipe stability in the seabed, that is, pipes with a submerged specific weight lower than half that of the soil will move up to the mudline. In turn, in all the wave conditions of these tests, pipe C sank almost the same net distance (9 to 11 cm). As was found in the other groups of tests, in a state of low pore pressure but with liquefaction, the soil loses loading capacity and the pipes sink. Taking into account the results obtained in the present research, and, as pointed out above, the specific weight (filler material) of the pipes also plays an important role in the interaction of the soil and the pipe.

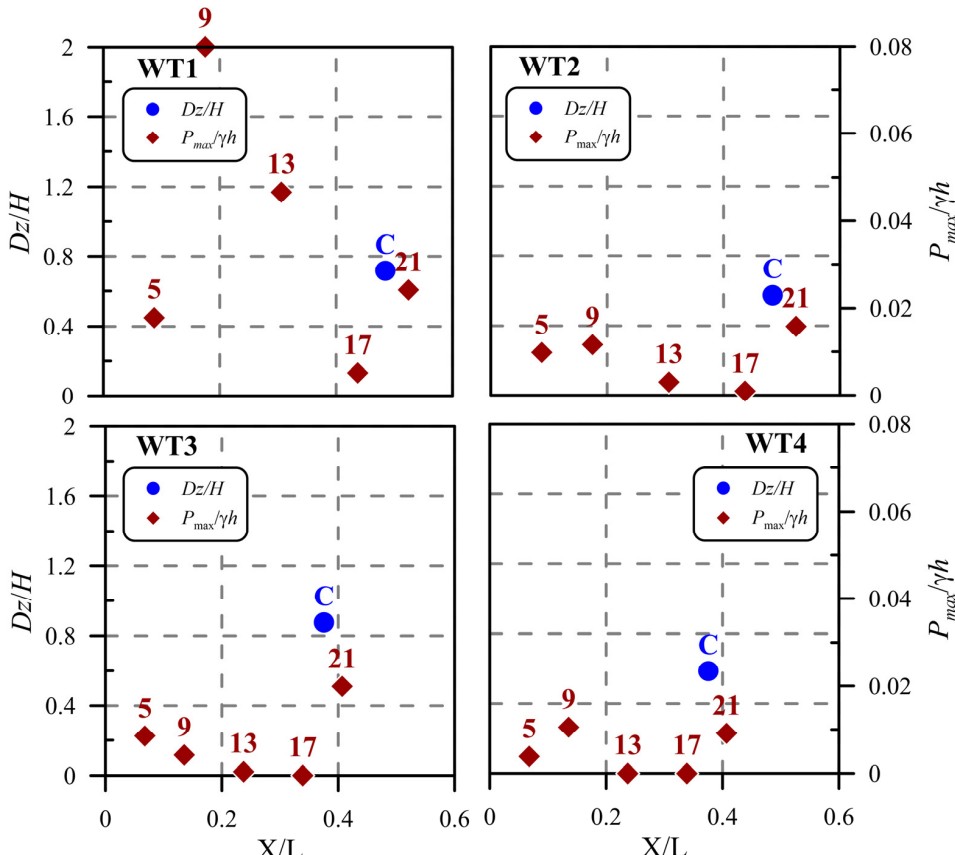

**Figure 15.** Vertical net displacements (final position–initial position) for pipes A, B and C (blue dots) and maximum pressure recorded by the transmitters in the line 9 cm below the mudline for all C4 tests.

## 4. Discussion

It is difficult to compare the results and findings obtained in the present research with those of previous works as there have been few experimental tests in which the soil failure was due to wave action. In most reported tests, the soil shows fluid-like behavior from the beginning of the tests [27], or the failure is due to the deadweight of the elements placed on the soil [15] or even without the effect of any structure or pipe on the soil [28–30]. Figure A15 shows evidence of the fluid-like behavior of the soil found in tests C1.

Teh et al., 2016 [13] found a pipe-seabed interaction similar to that in this research in terms of the fluid-like behavior of the soil in its upper layers which lets the pipe find an equilibrium sinking depth. In turn, the failure of the soil may induce flotation of the pipes as found by Magda et al. 2000 [31] and corroborated herein.

In spite of the unpredictability and uncertainty that persists, it was found that the effect of wave may influence the soil response greatly, and that the pore pressure tends to exhibit areas of accumulation, possibly related to soil anisotropy.

Water drainage from the soil mass, when this is loaded by wave action, seems to be the key factor in pipe stability. If an artificial mechanism to release pore pressure (i.e., wells or piles) can be found, the pipes may stay in place (as seen in the C3 tests).

## 5. Conclusions

Four groups of experimental tests were carried out, with the initial pipe positions and pipe densities varying, in order to investigate the occurrence and consequences of soil liquefaction around marine pipelines. Four wave conditions were tested for each group and the scaled model pipes, of different submerged specific weights, were placed at different depths. The most important findings for these particular tests are:

- When a significant pressure build-up process occurred (C1 and C3 tests), the maximum values were recorded by the transmitters located farthest from the wave maker (17 and 21);
- For pipes of similar specific weights, placed at the same distance from the wave maker (pipe B in tests C1 and C2) the pipe which was initially totally buried sank deeper in the mud than that which was initially half buried, even though the pressure was lower, as found by Vijaya kumar et al. [16]. However, the behavior of the pipes differed when the soil closer to the water-soil interface was more prone to liquefy (having a higher initial water content). In these circumstances, a pipe buried closer to the surface of the bed showed less vertical displacement, (pipe C in the C2 and C3 tests). The results of the present experimental study of muddy soils, show great unpredictability, due to the uncontrolled anisotropy of the soil, the inhomogeneity of the materials and the characteristics of the experiments related to the hand-made elements;
- It was found that a pipe with a submerged specific weight less than half that of the soil will move up. In addition, it is clear that the loss of loading capacity of the soil depends more on the fluid-like behavior of the soil than on the actual breaking of the soil matrix;
- As with any small-scale experiment involving sediment, this is a distorted experiment. The results and findings show a highly qualitative character. The goal of the work was to reproduce known field conditions and soil failure, in order to set the basis for further engineering recommendations and, in the near future, to test a solution.

**Author Contributions:** Conceptualization, E.M., R.S. and M.L.; methodology, M.G.N. and C.A.; validation, A.R., C.A. and M.G.N.; formal analysis, E.M. and M.G.N.; investigation, E.M., A.R. and C.A.; resources, R.S. and M.L.; data curation, E.M. and R.S.; writing—original draft preparation, E.M., M.G.N., A.R. and C.A.; writing—review and editing, R.S. and M.L.; visualization, E.M.; project administration, E.M. and M.G.N.; funding acquisition, R.S., M.L. and M.G.N. All authors have read and agreed to the published version of the manuscript.

**Funding:** This research was funded by AREDIS project from the European Regional Development Fund, NSRF—National strategic reference framework (2007–2013 Portugal) and the Regional operational program PORLISBOA, the Consejo Nacional de Ciencia y Tecnología (CONACYT) of Mexico and the Centro para el Desarrollo Tecnológico Industrial (CDTI) of Spain under contract C0004-2012-01 143095. The APC was funded by CONACYT-SENER/Sustentabilidad Energética through the Centro Mexicano de Inovación en Energías del Océano (CEMIE-Océano), grant number 249795.

**Data Availability Statement:** The data produced by this study are available upon request to the authors.

**Conflicts of Interest:** The authors declare no conflict of interest. The funders had no role in the design of the study; in the collection, analyses, or interpretation of data; in the writing of the manuscript, or in the decision to publish the results.

## Notation

| | |
|---|---|
| $D$ | pipe diameter (m) |
| $Dz$ | pipe sinking distance (m) |
| $D_{50}$ | value of the particle diameter at 50% in the cumulative distribution (m) |
| $e_{max}$ | maximum soil void ratio (-) |
| $e_{min}$ | minimum soil void ratio (-) |
| $H$ | wave height (m) |
| $h$ | water depth (m) |
| $KC$ | Keulegan-Carpenter number (-) |
| $k_0$ | coefficient of lateral earth pressure (-) |
| $L$ | wave length (m) |
| $P$ | pore pressure (m) |
| $Re$ | Reynolds number (-) |

$s_p$    pipe specific dead weight (kg/m$^3$)
$T$    period (s)
$z$    vertical coordinate (z = 0 mudline) (m)
$\gamma$    soil specific weight (kg/m$^3$)
$\rho_s$    soil mass density (kg/m$^3$)
$\rho_w$    water mass density (kg/m$^3$)
$\sigma'_0$    initial mean normal effective stress (kg/m$^2$)

**Appendix A**

In this section, various time series of pore pressures recorded for selected tests are presented. The depths indicated in the figures correspond to pressure transmitted locations, not to pipes' position.

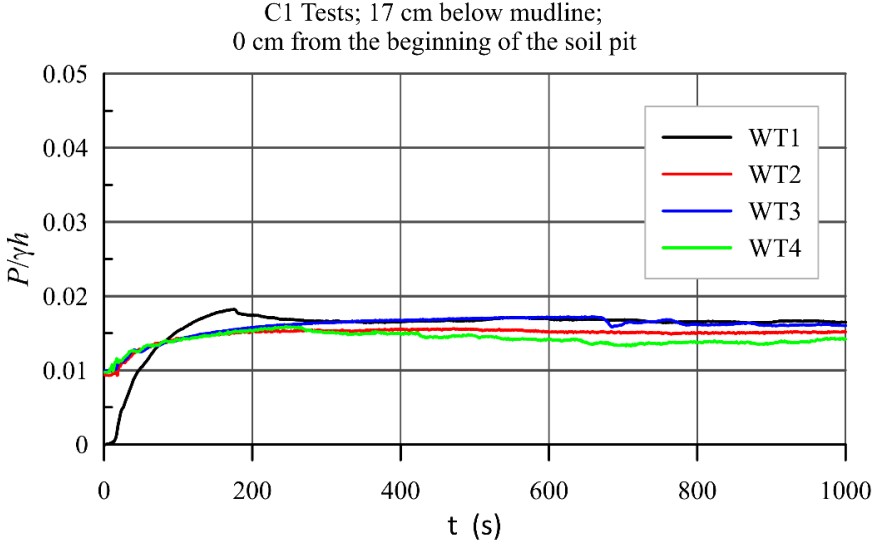

**Figure A1.** Phase averaged pressure for C1 Tests, 17 cm below mudline and 0 cm from the beginning of the soil pit.

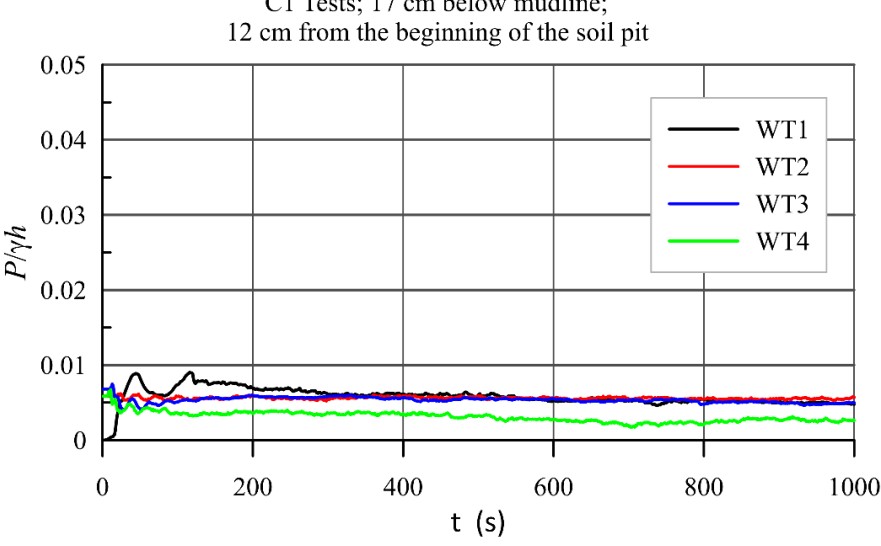

**Figure A2.** Phase averaged pressure for C1 Tests, 17 cm below mudline and 12 cm from the beginning of the soil pit.

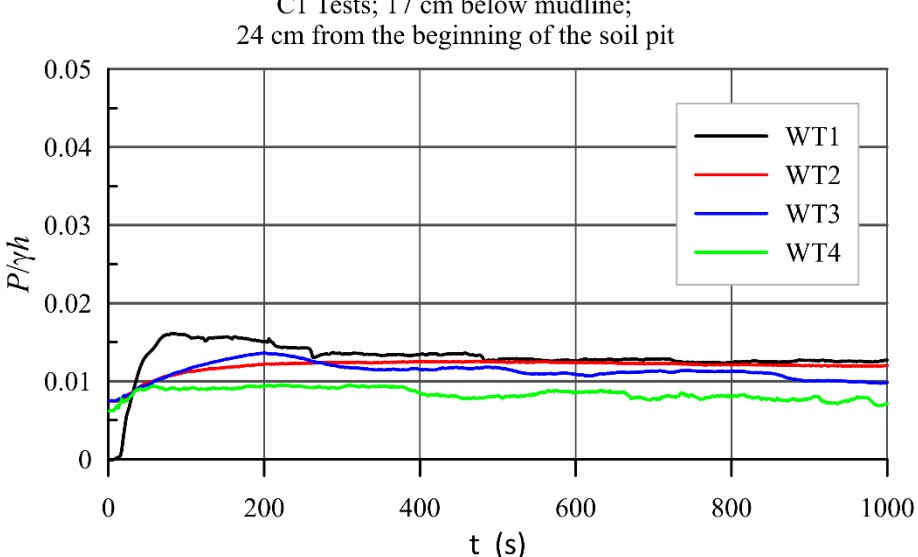

**Figure A3.** Phase averaged pressure for C1 Tests, 17 cm below mudline and 24 cm from the beginning of the soil pit.

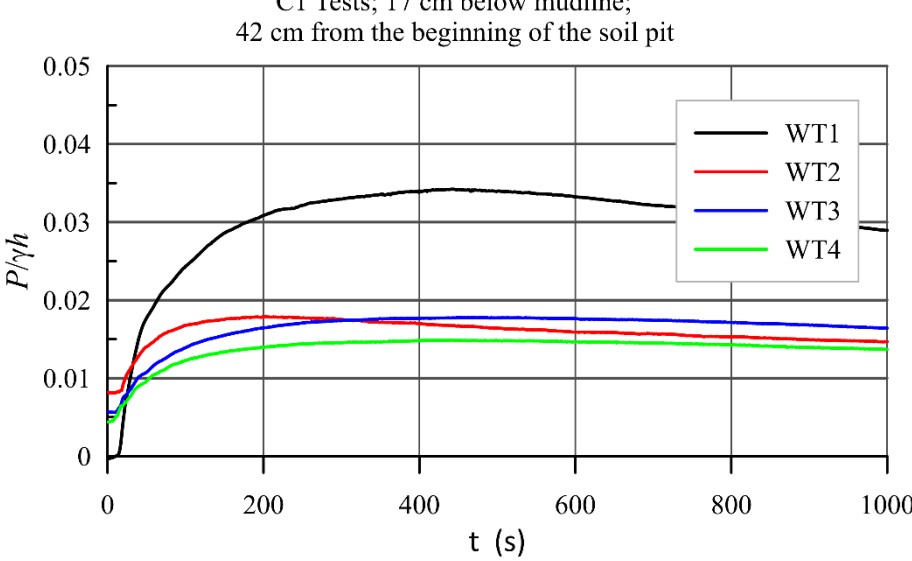

**Figure A4.** Phase averaged pressure for C1 Tests, 17 cm below mudline and 42 cm from the beginning of the soil pit.

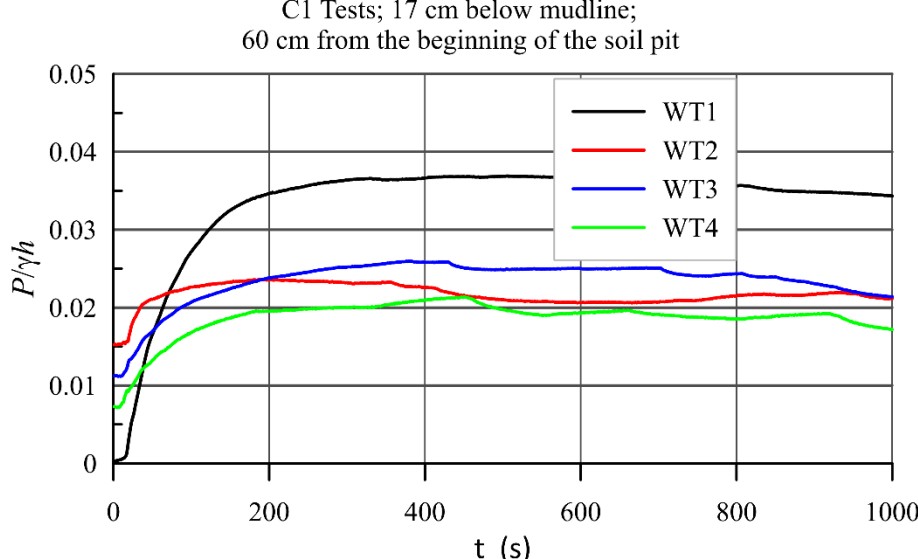

**Figure A5.** Phase averaged pressure for C1 Tests, 17 cm below mudline and 80 cm from the beginning of the soil pit.

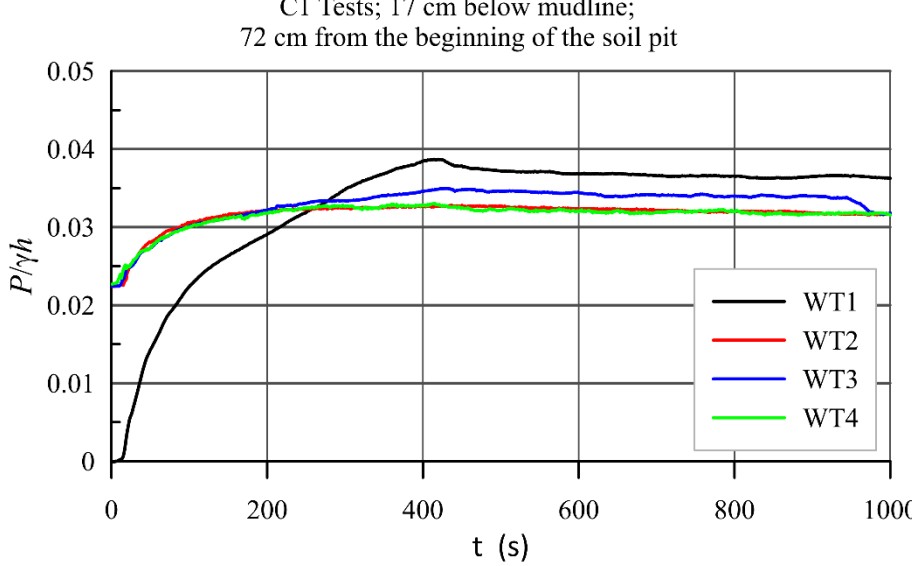

**Figure A6.** Phase averaged pressure for C1 Tests, 17 cm below mudline and 72 cm from the beginning of the soil pit.

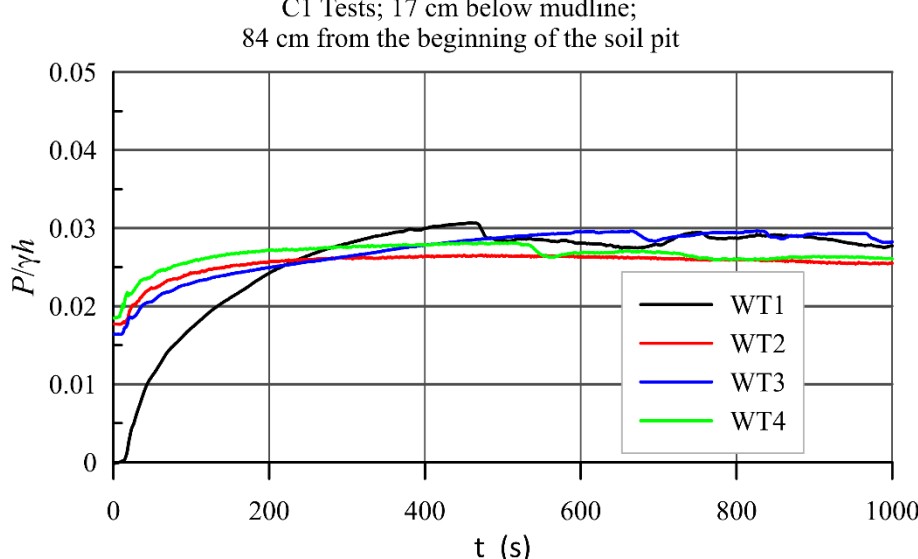

**Figure A7.** Phase averaged pressure for C1 Tests, 17 cm below mudline and 84 cm from the beginning of the soil pit.

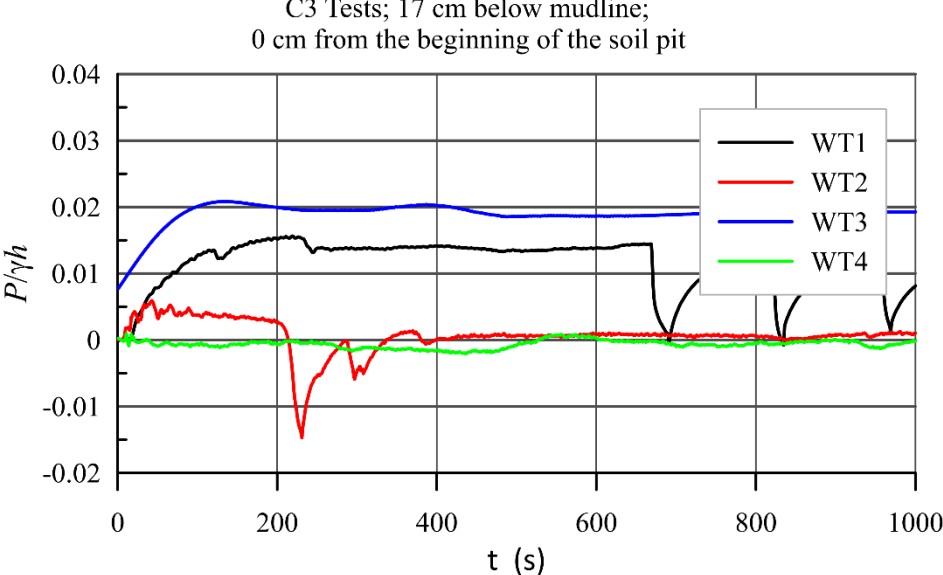

**Figure A8.** Phase averaged pressure for C3 Tests, 17 cm below mudline and 0 cm from the beginning of the soil pit.

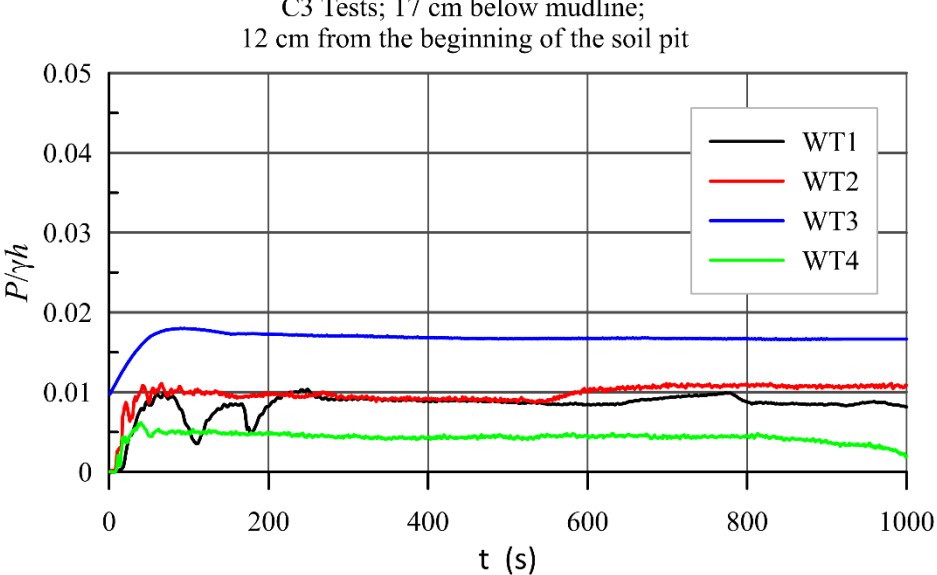

**Figure A9.** Phase averaged pressure for C3 Tests, 17 cm below mudline and 12 cm from the beginning of the soil pit.

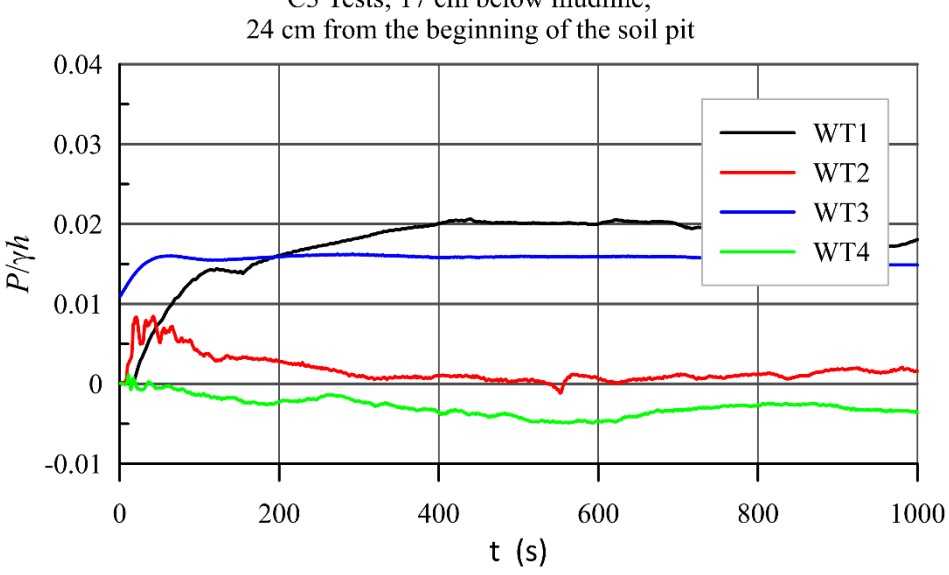

**Figure A10.** Phase averaged pressure for C3 Tests, 17 cm below mudline and 24 cm from the beginning of the soil pit.

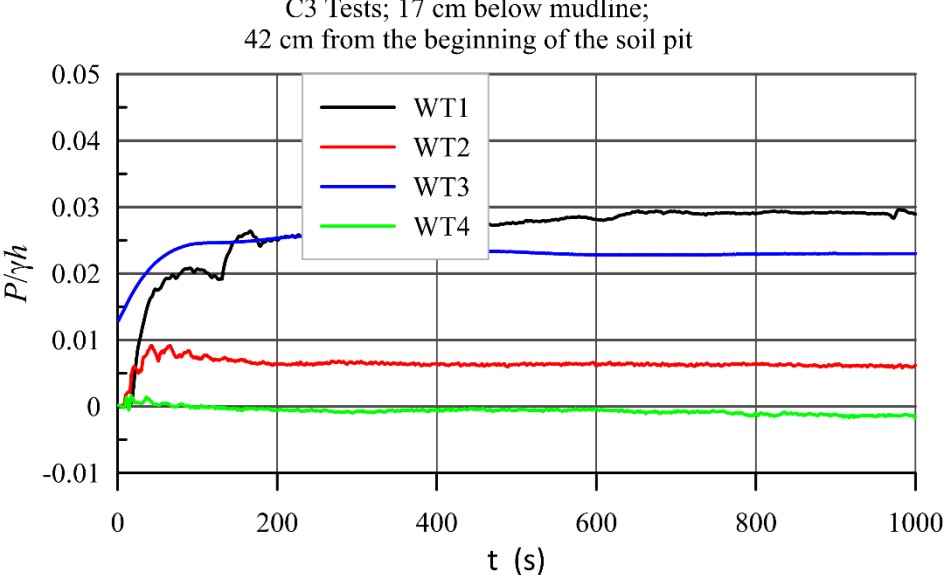

**Figure A11.** Phase averaged pressure for C3 Tests, 17 cm below mudline and 42 cm from the beginning of the soil pit.

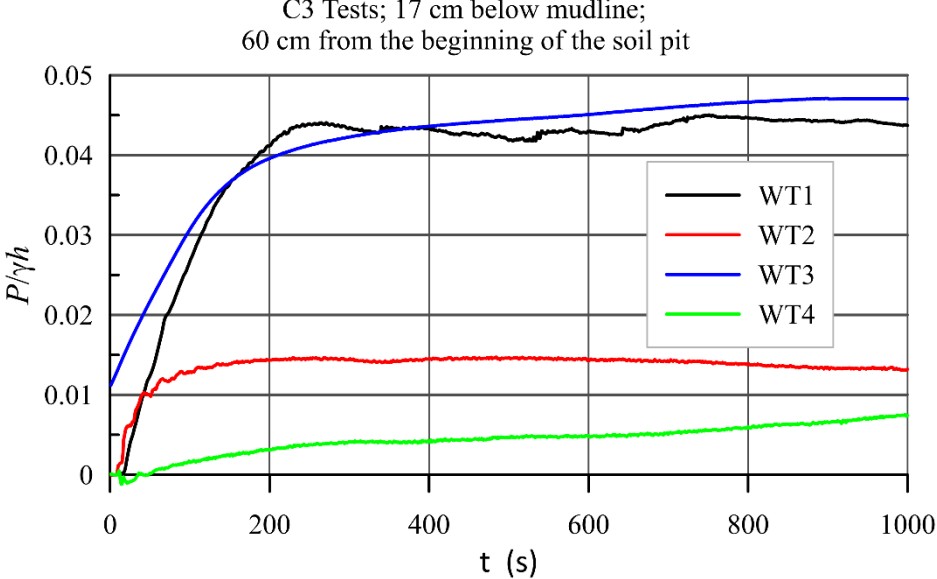

**Figure A12.** Phase averaged pressure for C3 Tests, 17 cm below mudline and 60 cm from the beginning of the soil pit.

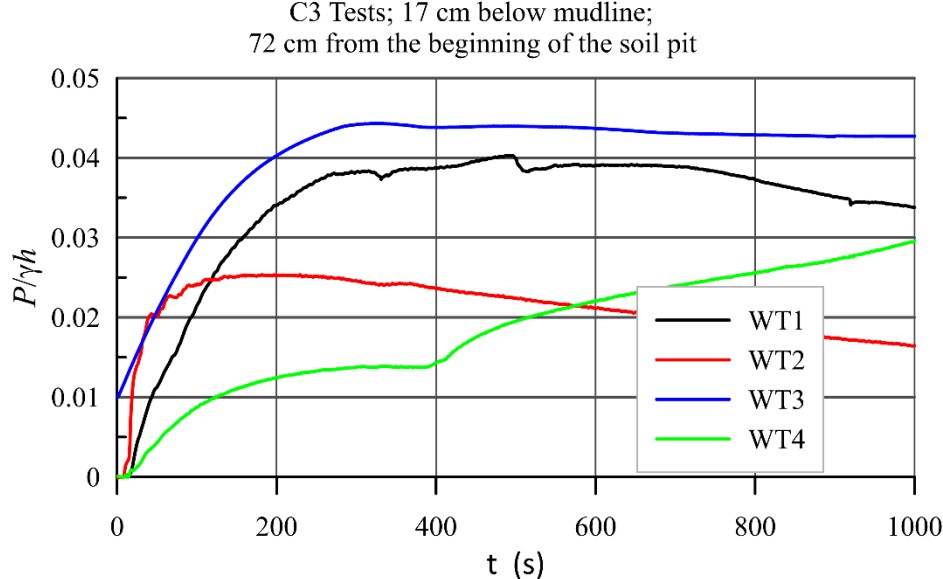

**Figure A13.** Phase averaged pressure for C3 Tests, 17 cm below mudline and 72 cm from the beginning of the soil pit.

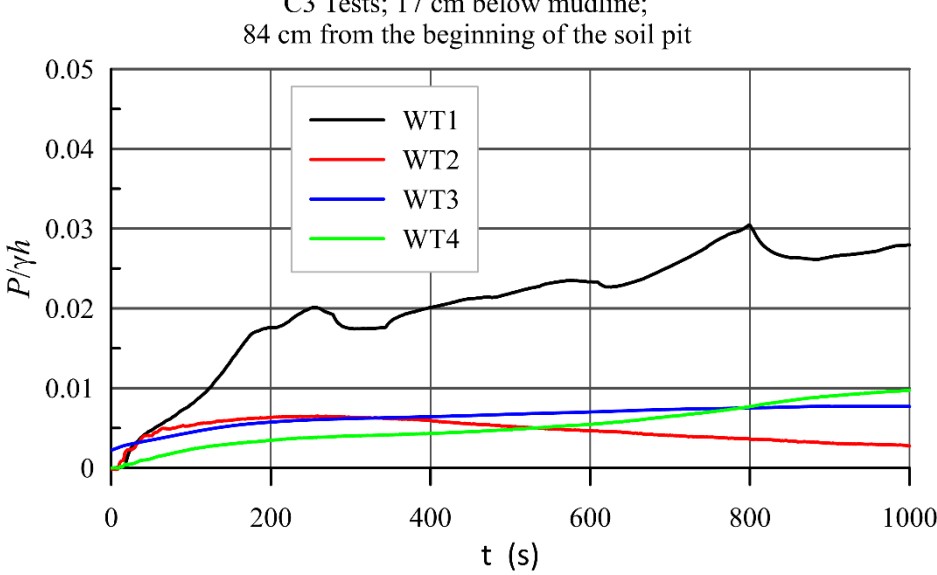

**Figure A14.** Phase averaged pressure for C3 Tests, 17 cm below mudline and 84 cm from the beginning of the soil pit.

Figure A15 shows a set of images taken during experiment C1 WT1. The initial position of the pipes can be seen (top images) and also photographs of the flume at the end of the tests, with water (panel c) and without water (panel d).

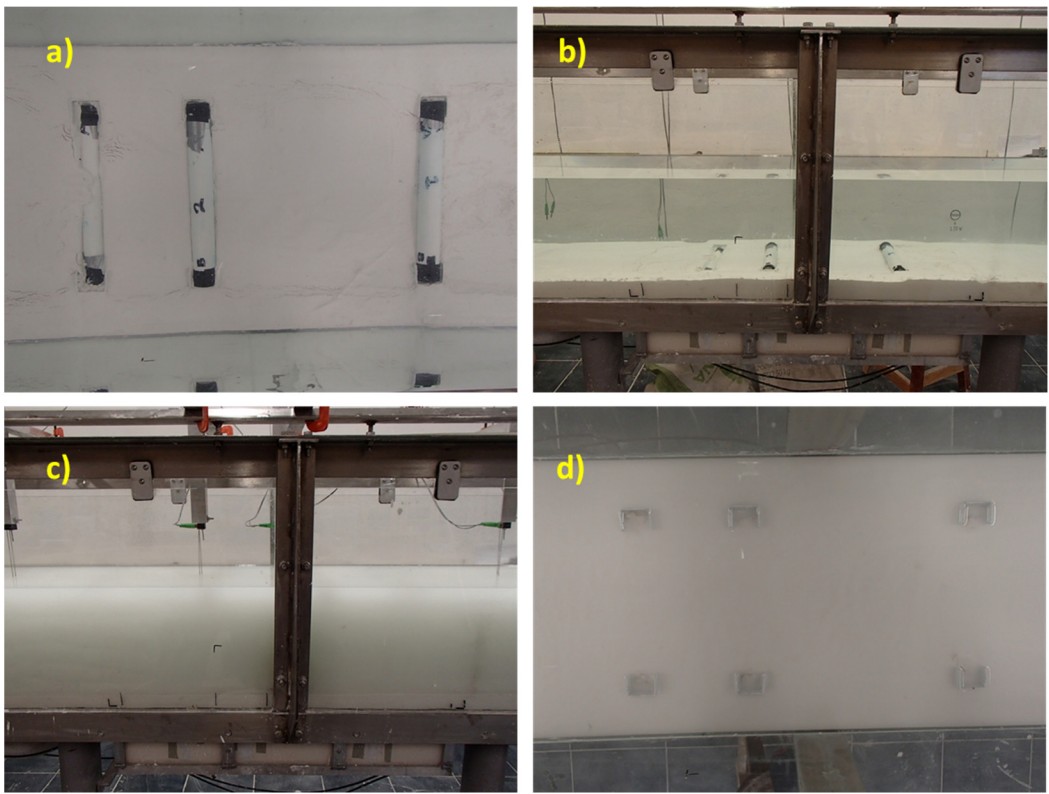

**Figure A15.** Photographs of C1 WT1 test (**a**); and (**b**) initial position of the pipes; (**c**) wave flume immediately after the test; and (**d**) wave flume without water.

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
