# Peer review of "Experiments on the Sinking of Marine Pipelines on Clayey Soils"

_water, doi:10.3390/w14050704_

Round 1
Reviewer 1 Report
The topic of this paper is very important in the research field of safe operation of pipes. This paper presents that an experimental study to investigate seabed-pipeline interactions with regard to soil liquefaction. The novelty of this paper is clear, especially to the safety assessment of the seabed-pipeline interactions. It is suitable for publication in this journal from the topic of this paper. I would suggest a moderate correction for this paper before it can be accepted for publication. The detailed comments listed below can be useful for the authors in improving their manuscript.
- This writing “[2] used the results from [7], to calibrate a simple, but workable modeling method.”is suggested to be modified.
- The undrained shear strength of marine clayey soil has a significant effect on pipe-soil interaction, especially for marine surface clayey soil. This paper does not mention the strength of clay sample. Is there any corresponding experimental data? It is suggested to add different sea area studies to support this the parameter design of the experiment. For examples: “Evaluation of undrained shear strength of surficial marine clays using ball penetration-based CFD modelling. Acta Geotechnica.”and “Centrifuge experiment on the penetration test for evaluating undrained strength of deep-sea surface soils. International Journal of Mining Science and Technology.”.
- According to paststudies (e.g., Scour below pipes in waves; Effect of pipeline surface roughness on peak impact forces caused by hydrodynamic submarine mudflow) , pipe surface roughness also has a significant influence. How does the authors consider the influence of this important factor?
- Deep-sea pipelines are generally laid flat on the seabed surface, which does not need to consider the influence of waves. However, shallow-water pipelines need to have a certain buried depth, which should consider the influence of waves. The setting of working conditions studied in this paper should be related to the actual situation.
- The thickness of the soil layer is a little thin. How is the boundary effect considered?
- The buried depths of the pipes in Table 2 are inconsistent with the Figures in the appendix, please check.
- The initial positions in Table 2 are suggested to be dimensionless.
- Whether the soil liquefaction process can be shown in the manuscript.
- The text in figures is too large, please resize it for publication.
- The ordinate of Figures in the appendix is not reasonable, please adjust it.
- Table 1 should be “Physical and mechanical propertiesof the soil mixtures”.
- It is suggested that Chapter 4 be divided into two parts, 4.Discussion and 5 Conclusions. 4. Discussion 4 is better to increase the comparison with previous studies.
Reviewer 2 Report
Overall, a good manuscript on an important topic. Before this manuscript should be published, however, the authors should comment on the following points:
- The authors consider cyclic action of the oscillatory regime exclusively due to waves (as it seems, wave – soil, and soil – pipeline interactions are uncoupled). However, in a real case, the pipeline may be subjected to vibrations by itself and thus contribute to liquefaction as well (wave/environmental loads – pipeline interaction, and then soil-pipeline interaction).How may findings of this study manifest themselves in the actual case where underwater currents and VIV also contribute to the liquefaction?
- For practical engineering applications of analyzing pipeline during its installation, to simplify the structural analysis of the pipe-lay, it is accepted to assume/adopt some simplified interaction rules between the soil and the pipeline (e.g., Trapper, P.A. 2020. Static analysis of offshore pipe-lay on flat inelastic seabed. Ocean Engineering 213: 107673). To which extent the finding of the present study can be adapted in that double-linear law? Should liquefaction be considered?
- How may ocean depth affect the results?
- How may the experimental setup be linked to the real field conditions? Are there expected side-effects?
Round 2
Reviewer 1 Report
It is suggested to accept this revised paper, as all my previous comments have been adequately addressed.
Reviewer 2 Report
The authors have addressed all the comments.
This manuscript is a resubmission of an earlier submission. The following is a list of the peer review reports and author responses from that submission.
Round 1
Reviewer 1 Report
In this study, a series of wave flume tests are performed to study the dynamic of pipeline with different buried depths induced by regular waves, especially the pore pressure accumulation in clay soils, and the sinking of pipeline are the focus. This reviewer initially carefully read this manuscript with a great enthusiasm. However, it is found that there are several serious problems on the results analysis, even though this kind of wave flume tests are interesting to some extent. Therefore, I can’t recommend this manuscript to be published at its current form.
Detail comments:
(1)The English writing needs to be polished to some extent. Overall, the taste of colloquial style in the writing is obvious. I suggest the writing is improved using formal and academic style. Additionally, some terminology are inappropriately used, e.g. soil’s load capacity, abrupt pressure etc.
(2)In the third paragraph of introduction, except for literature[1]、[2], there is another latest typical literature about the wave-induced sinking and floatation of pipeline (Dynamics of a pipeline buried in loosely deposited seabed to nonlinear wave & current), that should atract your attention.
(3)In the fifth paragraph of Introduction. It is not true that little was known about the wave- induced liquefaction. Actually, there were a number of meaningful works have been done, please see the literature review in Yang and Ye (2017, 2018) (Nonlinear standing wave-induced liquefaction in loosely deposited seabed, Wave & current-induced progressive liquefaction in loosely deposited seabed). In contrast, little affection has been paid on the seismic wave-induced soil liquefaction in the seabed foundation of marine structures, not the land structures. Please see Ye and Wang (2015) (Seismic dynamics of offshore breakwater on liquefiable seabed foundation).
(4)Overall, it is suggested that the content on the literature review could be slightly shorten. By the way, it is claimed that the uplift force on pipeline were measured. However, there is no the result presentation on this uplift force.
(5)Please rephrase the following: ① …placed in seven rows of 4 transducers…,② muddy soil was placed before and after the pit, … abrupt change in medium. By the way, please explain why it can avoid the contamination on the soil response, what meaning of abrupt change? Why it can steepen the wave?
(6)Water content should be expressed as percentage. Why the horizontal displacement was constrained in tests? Actually, the horizontal dynamics of pipeline could enhance the pressure build-up.
(7)In Table 3, there are totally 12 tests with different wave conditions. Why there is only one value for these parameters? What is the velocity and characteristic size when determining Re?
(8)This reviewer can not understand the meaning “cycle of lower pressure build-up followed by release at seen”? What means of abrupt pressure?
(9)Are you sure the location of the maximum wave height was found where the soil is expected to fail? Generally, the dynamics of pipeline and seabed soil to ocean wave is nonlinear. You can’t get this recognition adopting a linear thinking way. Additionally, why the wave height at WG 06 in WT 4 is the greatest? Where is the wave energy dissipation?
(11)In this study, the authors think that the soil becomes fail when the excess pore pressure is greater than its initial mean effective stress. It should be noticed that the above judgment standard is only applicable to the sandy bed without marine structure. In fact, due to the effect of soil cohesion, and the possible intensive soil-structure interaction, the soil could not become liquefied when Pexcess >= σ0 which has been clearly illustrated by Ye and He (2018), Ye and Wang (2015) (Seismic dynamics of offshore breakwater on liquefiable seabed foundation, Stability analysis of a composite breakwater at Yantai port, China: An application of FSSI-CAS-2D)..
(12)In Figure 6, the distributions of the maximum pore pressure are plotted. In this reviewer’s opinion, there is serious problem making the shown results is not reliable. There were totally 28 sensors buried in the soil. So only 28 data were obtained to plot Figure 6. The spatial resolution of the 28 data is not enough to plot a spatial distribution. Finally, the predicted zone of soil liquefaction in the pit is completely unreliable. It is not acceptable. In this reviewer’s opinion, it is impossible for the authors to judge the occurrence zone of soil liquefaction only depending on limited numbers of data of pore pressure.
Reviewer 2 Report
This paper presented an experimental study that investigates pipeline-seabed interactions. This study focused on verifying the theoretically computed areas of soil failure by analyzing the sinking depths of the pipelines. However, from a geotechnical view, soil failure is strongly related to the stress history of the soil sediment. For example, in the real ocean environment, the seabed (without a pipeline) has already been loaded with long-term wave action, altering the sediment strength or deformation. In this study, the sediments were consolidated for only a few hours to support the pipeline weight and consolidated for a further 24h before the beginning of the test. This preparation of seabed results in very soft seabed soil, which will continue to deform even with small amplitude wave loading. That means the soil in this study is too soft that can not represent the soil conditions in real seabed sediments. Furthermore, the position of the pipeline strongly depends on the bearing capacity of the seabed. The initial setup cannot represent the real pipeline embedment, because the system is actually not stable at the beginning of the experiments. The authors should carefully evaluate the influence of the initial conditions on the experimental results and discussion
Reviewer 3 Report
There are no comments on the article
Reviewer 4 Report
Dear Author
The authors of the article have examined the topic of experiments on the sinking of marine pipelines on clayey soils. This is a good topic and its authors have presented it well, but I need to ask a series of questions and comments for this article.
This article can be accepted and published in this journal after a minor review.
Some comments:
- In the introduction to line 34, briefly describe the two factors mentioned.
- Try not to use old references such as references 1 and 11.
- Article innovation is weak. Innovate to be rewritten.
- The dimensional analysis must be provided in the materials and methods section. There is no trace of dimensional analysis in this section. In the results and discussion section, the drawn diagrams should be based on dimensional analysis, and how to extract the dependent and independent parameters should be presented.
- Table 3 lists the dimensionless parameters but does not explain how to obtain these dimensionless parameters. Therefore, the article needs to provide a dimensional analysis.
- Parameters Dz/H and Pmax/γh, which are written on the diagrams in Figures 7, 9, 13, and 14, are not shown in Table 3. These parameters must be added between the independent parameters.
- You can use the following articles to perform dimensional analysis. I also recommend that you reference to these articles:
Daneshfaraz, R.; Aminvash, E.; Ghaderi, A.; Kuriqi, A.; Abraham, J. Three-Dimensional Investigation of Hydraulic Properties of Vertical Drop in the Presence of Step and Grid Dissipators. Symmetry 2021, 13, 895. https://doi.org/10.3390/sym13050895.
Daneshfaraz, R.; Ghaderi, A.; Sattariyan, M.; Alinejad, B.; Asl, M.M.; Di Francesco, S. Investigation of Local Scouring around Hydrodynamic and Circular Pile Groups under the Influence of River Material Harvesting Pits. Water 2021, 13, 2192. https://doi.org/10.3390/w13162192.
Daneshfaraz R, Aminvash E, Di Francesco S, Najibi A, Abraham J. Three-Dimensional Study of the Effect of Block Roughness Geometry on Inclined Drop. NMCE. 2021; 6 (1) :1-9
URL: http://nmce.kntu.ac.ir/article-1-359-en.html.
with the best wishes
Thank you